# Evaluation of Incipient Enamel Caries at Smooth Tooth Surfaces Using SS-OCT

**DOI:** 10.3390/ma15175947

**Published:** 2022-08-28

**Authors:** Yasushi Shimada, Takaaki Sato, Go Inoue, Hisaichi Nakagawa, Tomoko Tabata, Yuan Zhou, Noriko Hiraishi, Tadamu Gondo, Syunsuke Takano, Kei Ushijima, Hirotoshi Iwabuchi, Yukiko Tsuji, Sadr Alireza, Yasunori Sumi, Junji Tagami

**Affiliations:** 1Department of Cariology and Operative Dentistry, Graduate School of Medical and Dental Sciences, Tokyo Medical and Dental University, Tokyo 113-8549, Japan; 2Department of Periodontology, Shenzhen Stomatological Hospital, Southern Medical University, 1092 Jianshe Road, Luohu District, Shenzhen 518001, China; 3Biomimetics, Biomaterials, Biophotonics, Biomechanics & Technology (B4T) Laboratory, Department of Restorative Dentistry, University of Washington, Seattle, WA 98195, USA

**Keywords:** enamel, caries, demineralization, remineralization, optical coherence tomography, 3D image, diagnosis

## Abstract

(1) Background: Dental caries, if diagnosed at the initial stage, can be arrested and remineralized by a non-operative therapeutic approach preserving tooth structure. Accurate and reproducible diagnostic procedure is required for the successful management of incipient caries. The aim of this study was to evaluate the diagnostic accuracy of 3D swept-source optical coherence tomography (3D SS-OCT) for enamel caries at smooth tooth surface if the lesion was with remineralization. (2) Methods: Forty-seven tooth surfaces of 24 extracted human teeth visibly with/without enamel caries (ICDAS code 0–3) were selected and used in this study. The tooth surfaces of investigation site were cleaned and visually examined by four dentists. After the visual inspection, SS-OCT scanning was performed onto the enamel surfaces to construct a 3D image. The 2D tomographic images of the investigation site were chosen from the 3D dataset and dynamically displayed in video and evaluated by the examiners. A five-rank scale was used to score the level of enamel caries according to the following; 1: Intact enamel. 2: Noncavitated lesion with remineralization. 3: Superficial noncavitated lesion without remineralization. 4: Deep nonvacitated lesion without remineralization. 5: Enamel lesion with cavitation. Sensitivity and specificity for 3D OCT image and visual inspection were calculated. Diagnostic accuracy of each diagnostic method was calculated using weighted kappa. Statistical significance was defined at *p* = 0.05. (3) Results: 3D SS-OCT could clearly depict enamel caries at smooth tooth surface as a bright zone, based on the increased backscattering signal. It was noted that 3D SS-OCT showed higher sensitivity for the diagnosis of remineralized lesions and deep enamel lesions without cavitation, as well as cavitated enamel lesions (*p* < 0.05). No significant difference of specificity was observed between the two diagnostic methods (*p* > 0.05). Furthermore, 3D SS-OCT showed higher diagnostic accuracy than visual inspection (*p* < 0.05). (4) Conclusions: Within the limitations of this in vitro study, 3D SS-OCT showed higher diagnostic capacity for smooth surface enamel caries than visual inspection and could also discriminate lesion remineralization of enamel caries.

## 1. Introduction

Dental caries is a localized dissolution of tooth surface caused by metabolic events of cariogenic biofilm [1]. Caries is a dynamic process of demineralization and remineralization of the tooth, of which the fluctuation of equilibrium underneath the biofilm results in tooth demineralization and substantial breakdown [1,2].

In the early stage, dental caries can be prevented or arrested by simple and cost-effective interventions. The demineralization process of demineralized tooth surfaces without bacterial invasion or loss of structure can be reversed with proper plaque control and remineralization therapy [3]. Consequently, selection of remineralization strategies should be secured by the accurate diagnosis of caries in early stage. For enamel surface lesions, low-viscosity resinous infiltrant can also be an option to inhibit and arrest the carious progress [4]. Recent clinical study of resin infiltration for four years follow-up showed a significant decrease in the risk of caries progress [5]. Since remineralization therapy or resinous infiltrant for early caries does not involve any tooth grinding, it is less invasive and therefore advantageous for long-term oral function of the teeth.

Moreover, a risk of retreatment due to secondary caries often requires more extended or expensive interventions. This is because restoration replacement to control the secondary caries inevitably leads to progressive loss of tooth structure [6,7]. The cycle of invasive retreatment reduces the remaining tooth structure, and is detrimental to the tooth prognosis [8]. Therefore, detection of caries at an early stage with non-invasive or minimally invasive therapy is efficacious for preserving healthy dietary habits.

The International Caries Detection and Assessment System (ICDAS) has widely been used for diagnosis of caries because of its high validity and reliability [9]. ICDAS employs the change of optical properties for diagnosis of enamel demineralization by air drying. However, the lack of depth in information for enamel caries appears disadvantageous for the management of enamel caries.

Optical coherence tomography (OCT) is a noninvasive imaging technique to capture tomographic images of internal structure. [10] Previous studies demonstrated high accuracy of OCT for diagnosis of dental caries [11,12,13]. Polarization sensitive OCT (PS-OCT) can measure the phase and intensity images from the backscattered signal in two separate channels for imaging birefringent tissues [14]. Dental enamel is composed of enamel prisms made up of hydroxyapatite crystals, resulting in their being mostly birefringent due to the rod-like structures. Consequently, PS-OCT can image demineralization through the increase in backscattered intensity signals and birefringence changes [14].

In contrast, reflection of light occurs at the boundary surface of two objects with different refractive indices can effectively be utilized for the imaging of intensity-based non-polarized OCT, especially for the detection of interfacial defects of dental restorations and diagnosis of cavitation of caries [15]. Currently, swept-source OCT (SS-OCT) has been introduced in dentistry for diagnosis of caries and tooth cracks, as well as interfacial defects of resin composite restorations with improved imaging resolution [16,17,18]. Moreover, the high-speed scanning and processing capabilities of SS-OCT allow 3D imaging of tooth structure for diagnosis of dental diseases [19,20,21]. However, the diagnostic accuracy of 3D imaging of intensity-based OCT for incipient enamel caries is still limited. Intensity-based OCT does not have the capability to measure the birefringence changes of enamel, which seems to influence the diagnostic accuracy for incipient enamel caries.

Therefore, the aim of this study was to evaluate the capability of intensity-based 3D OCT for diagnosis of incipient enamel caries at smooth surface ex vivo. We hypothesized (H0) that 3D images of intensity-based OCT would have no improvement on diagnosis for early enamel caries compared with visual inspection based on the ICDAS system. This null hypothesis was tested against the alternative hypothesis (H1) of difference.

## 2. Materials and Methods

### 2.1. Study Design

Ethical approval for this study was obtained from the Institutional Review Board of Tokyo Medical and Dental University (approval number D2013-013) in accordance with the guidelines, Ethical Guidelines for Medical and Health Research involving Human Subjects. Twenty-four extracted human molars stored at 4 °C in water were used in this study. The usage of the teeth was approved by the Institutional Review Board of Tokyo Medical and Dental University (approval number D2013-022).

The proximal surfaces of the teeth were visually inspected with 3× magnification loupes and classified using ICDAS II. Both caries-free enamel surfaces (code 0) and enamel surfaces with demineralization (codes 1 to 3), in a total of 47 smooth surfaces (ICDAS code 0: 17 surfaces, code 1: 17 surfaces, code 2: 8 surfaces, code 3: 5 surfaces) of 24 molars were selected for this study.

After the removal of calculus or debris, digital photographs of the surfaces were obtained under air-dried conditions without magnification.

### 2.2. Visual Inspection

Visual inspection for the selected enamel surfaces was performed by four examiners, each with 2 years of clinical experience in cariology and operative dentistry, who participated in this study. The examiners were the first-time viewers for OCT images. The presence or absence of enamel demineralization or caries was recorded and scored as follows:

Score 0:No caries or demineralization of enamel.Score 1:Shallow enamel demineralization. Enamel demineralization appears at the outer half of the enamel thickness.Score 2:Deep enamel demineralization. Enamel demineralization appears to penetrate into the inner half of the enamel.Score 3:Enamel demineralization with remineralization.Score 4:Enamel caries with cavitated surface.

### 2.3. SS-OCT System

The OCT system employed in this study was swept-source OCT (Figure 1). The central wavelength of incident light was 1310 nm, and the wavelength sweeps ranged from 1240 to 1380 nm with 50-kHz sweep rate. This SS-OCT system can acquire complete tomographic images of a volume (10 mm long × 10 mm wide × 8 mm deep of optical pass depth) in 3.4 s.

The power of the sample beam was 18 mW, and the system had a sensitivity of 100 dB. The optical resolution of the 3D data set in air was less than 11 µm in depth and 40 µm in lateral and axial dimensions.

The 3D OCT images were acquired by scanning the smooth enamel surfaces. The teeth were kept moist during the acquisition of 3D OCT images. Backscattered light from the object was coupled back to the system, digitized over a time scale, and analyzed by Fourier transformation to reveal the depth information of the sample. Cross-sectional images were generated by a raster scanning of light achieved by galvanometer-based scanner. A sequence of cross-sectional images obtained by the raster scanning enabled the generation of 3D volumetric datasets, thereby yielding 3D images.

### 2.4. Evaluation of Enamel Lesions Using SS-OCT

For image evaluation, an LCD monitor was used to display the SS-OCT images. The 3D images of smooth enamel surfaces, as well as the sequence of two-dimensional (2D) tomographic images extracted from the 3D dataset, were dynamically displayed in video format using a custom-developed software. The display settings, such as brightness and contrast, were kept the same for all images and examiners.

In order to reach a consensus regarding the diagnostic criteria, the principal evaluator (YS) discussed SS-OCT imaging with the 4 dentists in a 1-h session. Each examiner then scored the level of caries on the basis of visual inspection and 3D OCT images separately. The following 5-point rank scale was used to score the level of caries progression.

Score 0:No caries or demineralization of enamel. In OCT, the obtained signal was the same level as that for the surrounding normal enamel.Score 1:Shallow enamel demineralization. In OCT, the signal intensity within the outer half of enamel thickness was enhanced with no enamel surface loss.Score 2:Deep enamel demineralization. In OCT, the signal intensity within the inner half of enamel thickness was enhanced with no enamel surface loss.Score 3:Subsurface enamel demineralization with remineralization. In OCT, the signal intensity of inner enamel was enhanced but signal intensity of superficial enamel was the same level as the surrounding intact enamel. Loss of enamel was not observed.Score 4:Enamel caries with cavitated surface.

### 2.5. Histological Observation of Enamel Lesion

The diagnostic observation of enamel lesions was validated by observations of histologically sectioned teeth using confocal laser scanning microscopy (CLSM; 1LM21H/W, Lasertec Co., Yokohama, Japan). The teeth were trimmed horizontally from the occlusal buccal surface using a highspeed rotated diamond stone and wet silicon carbide papers under running water, followed by further polishing with diamond paste down to 3 μm. The cross-sectioned polished surface was at the center of the lesion. The polished specimens were ultrasonically cleaned for 3 min and were observed using CLSM at magnification of 125×.

### 2.6. Statistical Analysis

Statistical analyses were performed using a statistical software package (SPSS-2 for Windows, SPSS, Chicago, IL, USA). Indices of sensitivity and specificity were calculated for the results of SS-OCT and visual inspection. For sensitivity, detection of superficial enamel demineralization, deep enamel demineralization, remineralized enamel lesion, and cavitated enamel caries were calculated. Receiver operating characteristic (ROC) analysis was performed to calculate the area under the ROC curve (AUC). Diagnostic accuracies of the 3D OCT images and visual inspections were also calculated by agreement with histological findings using weighted kappa. The results obtained by the 4 examiners were compared non-parametrically using the Mann-Whitney U test at a significance level of α = 0.05.

## 3. Results

In this study, 3D OCT could depict the enamel demineralization as the area of increased brightness compared with intact enamel. Representative images are shown in Figure 2, Figure 3, Figure 4, Figure 5, Figure 6 and Figure 7. Video image of dynamic slicing from the 3D OCT dataset are presented in Appendix A. The sensitivity, specificity, and AUC values of 3D OCT and visual inspection based on the validation are shown in Table 1, Table 2 and Table 3. Agreements of 3D OCT and visual inspection with histology are shown in Table 4.

The 3D OCT showed higher sensitivity for detection of deep enamel demineralization, enamel lesion with surface remineralization, and enamel caries (Mann-Whitney U test, *p* < 0.05). The 3D OCT also showed higher values of AUC and agreement with histology than the visual inspection (Mann-Whitney U test, *p* < 0.05). With regards to specificity, no significant difference was observed between the two diagnostic methods (Mann-Whitney U test, *p* > 0.05).

## 4. Discussion

OCT imaging of enamel demineralization has been reported to show increased brightness with enhanced backscattered signal intensity at the lesion site compared with intact enamel [22]. In this study, gray scale SS-OCT images showed demineralized enamel as a white zone with the information of lesion penetration depth (Figure 4). Consequently, higher sensitivity to most of the lesion thresholds employed in this experiment could be obtained by SS-OCT (Table 1). Especially, deep enamel lesion could easily be determined and discriminated from shallow enamel demineralization, since the near infrared laser employed in SS-OCT can penetrate the whole thickness of enamel and underlining dentin to construct a cross-sectional image beyond the dentin–enamel junction (DEJ).

Moreover, the enhanced scanning and processing speed of SS-OCT allows the construction of 3D images under clinical situations. So far, we can diagnose the lesion location and penetration depth even at the blind area accurately in three dimensions (Table 1, Table 2, Table 3 and Table 4). These beneficial imaging characteristics in SS-OCT facilitate the selection of a less invasive and more acceptable approach for detecting enamel demineralization.

Demineralization and remineralization processes of enamel caries are known to be closely related to the variation of birefringence derived from enamel prisms and apatite crystal structures. PS-OCT, which incorporates scanning images from two orthogonal directions, has the capability to image tissue birefringence and has been employed for imaging tooth demineralization [14,23]. Baumgartner et al. evaluated the increase in backscattered signals and phase differences of caries lesions using PS-OCT [14]. Fried et al. employed a 1310 nm light source for PS-OCT and reported that the reflected signal from the enamel surface layer depended on the angle of incident light and that the intensity of the surface signal depended on the phase difference [23]. The SS-OCT employed in this study was an intensity based set up, which did not have the capability to discriminate birefringent changes of enamel demineralization. However, high diagnostic accuracy for caries, due to its high image resolution and improved image depth of SS-OCT, have been reported both in vitro and in vivo experiments [15,19]. Reflection and scattering of light as a phenomenon results in Fresnel reflection that occurs at the interface between two media having different refractive indices.

Hariri et al. evaluated the refractive indices of demineralized enamel and dentin using SS-OCT by measuring the path length difference after the demineralization, and compared the results with mineral density [24]. The results indicated significant correlation between the refractive indices and mineral density. Consequently, intensity-based SS-OCT can image tooth demineralization as a bright zone. Despite the fact that we did not measure the mineral density of enamel demineralization, high diagnostic accuracy of SS-OCT for enamel demineralization was observed (Table 1, Table 2 and Table 3). The increase in brightness of demineralization in SS-OCT images was due to the changes of the refractive index resulting from formation of numerous submicron-size defects within the enamel lesion. Therefore, if the increased brightness of the SS-OCT image correlates with the severity of enamel demineralization, it may be utilized for the quantitative assessment of enamel demineralization. However, previous study has reported that demineralized enamel between 30% to 70 % mineral density did not show significant difference of scattering intensity in PS-OCT [22]. The interaction of SS-OCT images with changes in optical properties of enamel needs further detailed studies.

Along with diagnosing enamel demineralization, we tried to evaluate the presence of remineralization within the SS-OCT images of enamel caries (Figure 5, Figure 6 and Figure 7). Jones et al. reported that reflectivity increased in enamel demineralization and decreased after the remineralization to become close to the level of healthy enamel [22]. Our SS-OCT setup revealed some of the enamel lasions forming a surface layer with a brightness similar to the level of surrounding intact enamel. Moreover, OCT image brightness of subsurface lesions forming on the surface layer was lower than the demineralization without the surface layer (Figure 7). Consequently, we speculated this type of enamel lesion was remineralized and at the chronic stage. Observation of enamel demineralization using cross-polarization OCT (CP-OCT), which also has the function to construct images without the influence of tissue birefringence, reported that remineralization could be identified in enamel demineralization [25]. Similar results were observed in this study, as SS-OCT could identify the formation of a surface remineralization layer within the enamel demineralization (Figure 5). Hariri et al. remineralized pre-demineralized enamel and dentin, and measured the changes of refractive index using SS-OCT. They found the refractive index after the remineralization was close to the sound tooth [24]. It is highly probable that the effect of remineralization therapy on enamel demineralization can be monitored to manage lesion extent and depth using SS-OCT.

In this study, the null hypothesis (H0) was rejected and the alternative hypothesis (H1) was accepted. SS-OCT showed high sensitivity and AUC for the diagnosis of incipient enamel caries localized at smooth tooth surfaces. All the examiners recruited in this study had only two years of clinical experience, and had no experience of OCT images to evaluate caries. Consequently, we needed to give them a 1-h session for OCT imaging of caries before the evaluation of SS-OCT images. Since OCT images are different from radiographs, it was necessary for our examiners to have knowledge of the imaging characteristics of OCT in order to diagnose caries. However, our results showed a high diagnostic accuracy, which is likely to be improved with further experience [12].

Our study has several limitations which need to be disclosed. We observed natural enamel demineralization in extracted teeth preserved in solution. Since the optical property of extracted teeth may differ from intraoral teeth, an in vivo study is necessary using SS-OCT. Moreover, correlation between SS-OCT image and mineral density of the lesion site was not evaluated in this study. Further studies are necessary to utilize SS-OCT in clinical situations for diagnosis of enamel demineralization and remineralization.

## 5. Conclusions

Within the limitations of this study, 3D SS-OCT could visualize smooth surface enamel caries by increased brightness with the information of lesion location and lesion depth. The 3D OCT showed higher diagnostic accuracy than visual inspection, with the capability to image the state of remineralization in enamel caries. SS-OCT may possibly be used to monitor the effect of remineralization therapy on enamel caries and may potentially provide the lesion activity.

## Figures and Tables

**Figure 1 materials-15-05947-f001:**
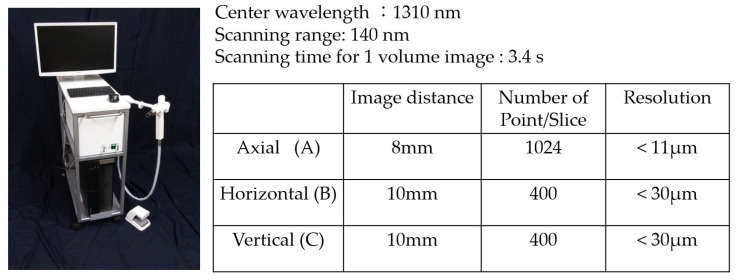
SS-OCT system used (Prototype Yoshida Dental OCT).

**Figure 2 materials-15-05947-f002:**
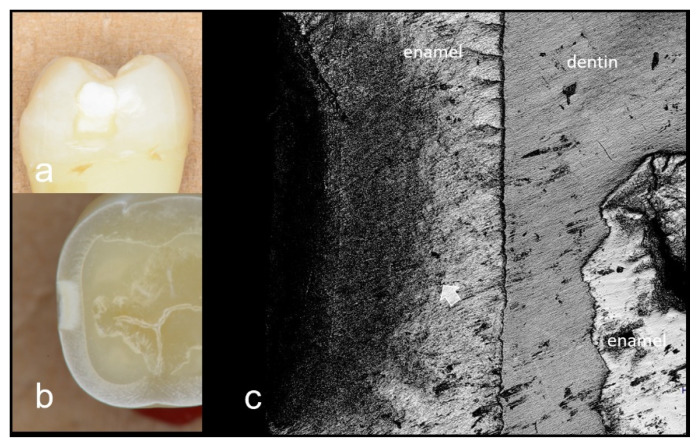
Images of deep enamel demineralization (score 2). (**a**) Front view of enamel demineralization; (**b**) Histological view of the enamel demineralization sectioned horizontally; (**c**) CLSM image. Morphological change within the demineralization is evident. Enamel demineralization penetrates inner half of enamel (white arrow).

**Figure 3 materials-15-05947-f003:**
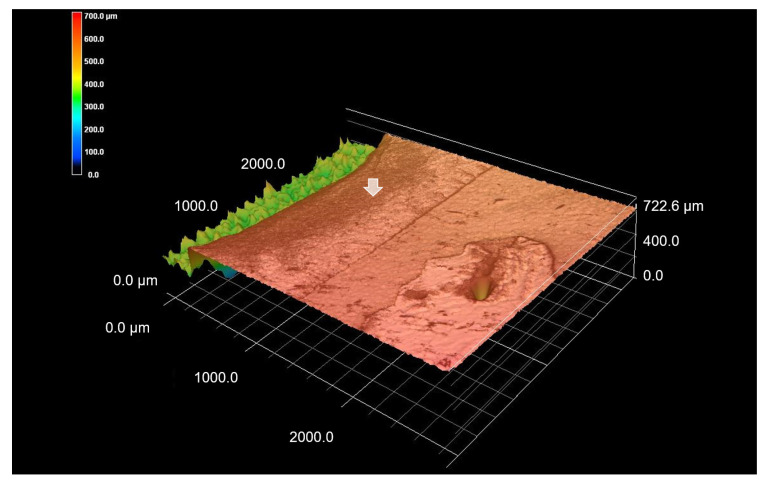
3D CLSM images of deep enamel demineralization (score 2). 3D image of Figure 2c. Zone of demineralization is clearly imaged as loss of surface (white arrow).

**Figure 4 materials-15-05947-f004:**
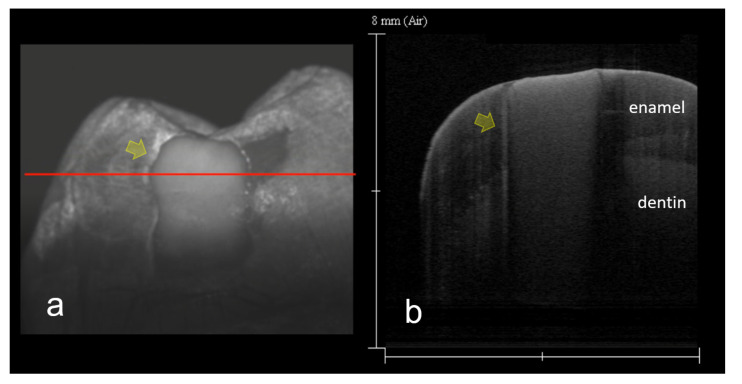
SS-OCT Images of deep enamel demineralization (score 2). Same tooth as Figure 2. (**a**) 3D SS-OCT image. Due to the scattering from the internal structure, enamel demineralization is clearly shown as a white zone (arrow); (**b**) 2D SS-OCT image chosen from the 3D image (**a**) along red line. Demineralization penetrates enamel inner zone (arrow). Dynamic slicing 3D video is presented in Appendix A.

**Figure 5 materials-15-05947-f005:**
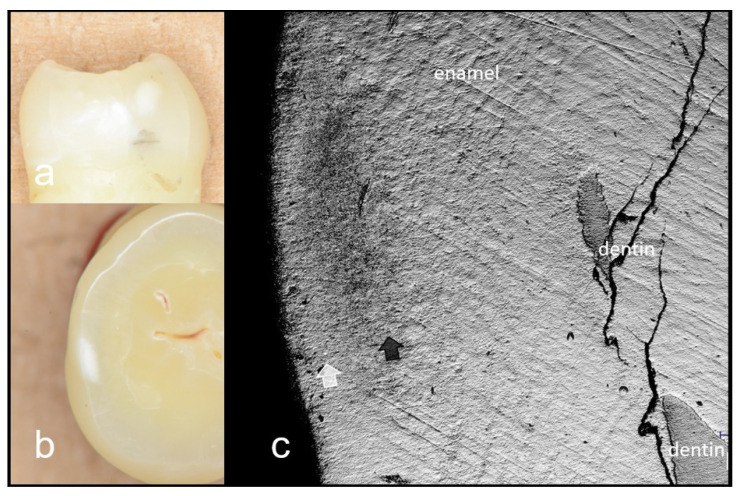
Images of remineralized enamel demineralization (score 3). (**a**) Front view of enamel demineralization; (**b**) Histological view of enamel lesion sectioned horizontally; (**c**) CLSM image. Morphological change within the lesion is not so evident (black arrow). Surface enamel of the lesion morphologically continues with surrounding enamel, suggesting the remineralization (white arrow).

**Figure 6 materials-15-05947-f006:**
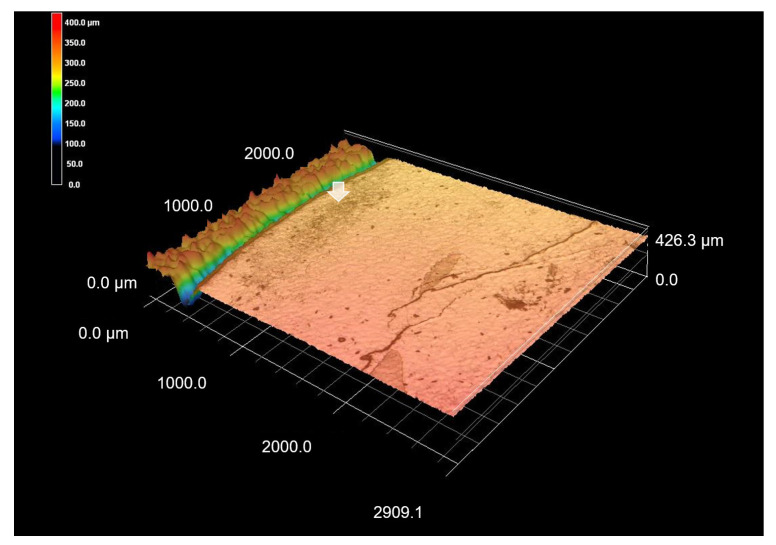
3D CLSM images of remineralized enamel demineralization (score 3). 3D image of Figure 5c. Zone of demineralization is not evident (white arrow).

**Figure 7 materials-15-05947-f007:**
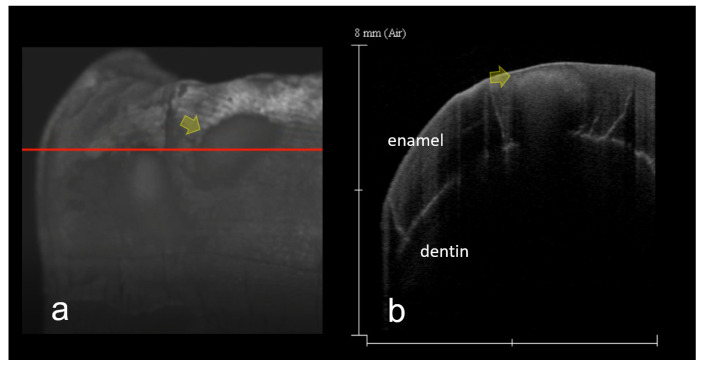
SS-OCT images of remineralized enamel demineralization (score 3). Same tooth as Figure 3. (**a**) 3D SS-OCT image of remineralized enamel demineralization. Due to the scattering from the subsurface lesion, presence of demineralization is imaged (arrow); (**b**) 2D SS-OCT image chosen from the 3D image (**a**) along red line. Formation of remineralization zone of lesion surface can be observed (arrow). Dynamic slicing 3D video is presented in Appendix A.

**Table 1 materials-15-05947-t001:** Sensitivity of visual inspection and SS-OCT.

	Visual Inspection	SS-OCT	Significance
shallow demineralization	0.67	0.78	NS
deep demineralization	0.15	0.65	S
remineralization	0.036	0.69	S
cavitation	0.42	0.83	S

Significance between visual inspection and SS-OCT. S: significant, NS: not significant (Mann-Whitney U test, α = 0.05).

**Table 2 materials-15-05947-t002:** Specificity of visual inspection and SS-OCT.

Visual Inspection	SS-OCT	Significance
0.85	0.88	NS

Significance between visual inspection and SS-OCT. S: significant, NS: not significant (Mann-Whitney U test, α = 0.05).

**Table 3 materials-15-05947-t003:** AUC (ROC analysis) of visual inspection and SS-OCT.

Visual Inspection	SS-OCT	Significance
0.83	0.97	S

Significance between visual inspection and SS-OCT. S: significant, NS: not significant (Mann-Whitney U test, α = 0.05).

**Table 4 materials-15-05947-t004:** Agreement of visual inspection and SS-OCT with histological observation.

Visual Inspection	SS-OCT	Significance
0.52	0.77	S

Significance between visual inspection and SS-OCT. S: significant, NS: not significant (Mann-Whitney U test, α = 0.05).

## Data Availability

Not applicable.

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
