# Peer review of "Evaluation of Incipient Enamel Caries at Smooth Tooth Surfaces Using SS-OCT"

_materials, 2022, doi:10.3390/ma15175947_

Round 1

Reviewer 1 Report

The author Shimada et al. utilized the SS-OCT technology for the early detection of enamel caries on tooth surfaces. This work is certainly an interesting approach to the early detection of the caries problems that many patients suffer today. The work focuses on the procedures and the viability of the SS-OCT detection technology. It appears that based on the imaging, the increase in brightness and contrast (from the backscattering) suggests the location of the lesion site on the tooth. The results suggest a significant improvement in detection and conclusion over naked eye inspection. The author also reasonably points out that in vivo is necessary since every tooth in intraoral condition may differ and this requires further investigation. In my opinion, this work is suitable for the MDPI Materials scope and can be published after some comments are addressed.

  1. Have you considered analyzing the 2D image and extracting a quantifiable value? For example, if the brightness can be correlated with the severity of caries. Then the intensity of the brightness can be assigned to a colorimetric/intensity value. This can potentially further improve the system in a point-of-care direction.

Author Response

We wish to express our appreciation to the Reviewers for insightful comments on our paper. The comments have helped us significantly improve the paper.

Response to Reviewer 1

Comment 1.Have you considered analyzing the 2D image and extracting a quantifiable value? For example, if the brightness can be correlated with the severity of caries. Then the intensity of the brightness can be assigned to a colorimetric/intensity value. This can potentially further improve the system in a point-of-care direction.

 Response: Thank you for constructive feedback. We modified Discussion 4th paragraph to describe the possibility of quantitative analysis of enamel demineralization using SS-OCT image.

Reviewer 2 Report

would appreciate the author working on evaluation of diagnostic accuracy of 3D swept-source optical coherence tomography (3D SS-OCT) for enamel caries at smooth tooth surface if the lesion was with remineralization. The methodology and results are appropriate. However, the following suggestions to improve the readability and more clarity. 

·       Repetition of caries word in the title. It should be like- smooth tooth surfaces.          Line no.2

·       Keywords not defined.                                                                 Line no. 3

·       Word “PS-OCT” not defined previously                                       Line no. 63

·       Word “SS-OCT” not defined previously                                       Line no. 63

·       Mann-whitney U-test not explained anywhere in the article        Line no. 169

·       What is “DEJ”?                                                                            Line no. 223 

·       “Polarization-sensitive (PS) OCT “, need to defined previously. Line no. 242

·       Word “CP-OCT” not defined                                                       Line no. 264

·       No Acknowledgement statement                                                Line no. 299

Author Response

We wish to express our appreciation to the Reviewers for insightful comments on our paper. The comments have helped us significantly improve the paper.

Response to Reviewer 2

We wish to express our deep appreciation to the reviewer for educational and detailed comments.

Comment 1. Repetition of caries word in the title. It should be like- smooth tooth surfaces. Line no.2

Response: Thank you for pointing out. We modified the title.

Comment 2. Keywords not defined. Line no. 3

Response: Thank you for pointing out. We added keywords.

Comment 3. Word “PS-OCT” not defined previously Line no. 63

Response: Thank you for pointing out. We spelled out “PS-OCT”.

Comment 4. Word “SS-OCT” not defined previously Line no. 63

Response: Thank you for pointing out. We spelled out “SS-OCT”.

Comment 5. Mann-Whitney U-test not explained anywhere in the article Line no. 169

Response: Thank you for pointing out. We showed the results of Mann-Whitney U-test in Results 2nd paragraph and Tables 1,2,3,and 4.

Comment 6. What is “DEJ”? Line no. 223

Response: Thank you for pointing out. We spelled out “DEJ”.

Comment 7.

“Polarization-sensitive (PS) OCT “, need to defined previously. Line no. 242

Response: Thank you for pointing out. We spelled out “PS-OCT” in Introduction 5th paragraph.

Comment 8. Word “CP-OCT” not defined Line no. 264

Response: Thank you for pointing out. We spelled out “CP-OCT”

Comment 9. No Acknowledgement statement Line no. 299

Response: Thank you for pointing out. We added our Acknowledgement.

Reviewer 3 Report

Dear Authors,

The main question address by this manuscript is evaluate the use of SS-OCT to improve the diagnostic of enamel caries lesions.

Very interesting topic since prevention and monitoring of incipient caries lesions is as import clinical parameter to control the caries risk of patients.

This paper is unique since SS-OCT was never used for caries diagnosis before.

This add to the areas of cariology and restorative dentistry showing the potential of this tool to assist clinicians in detecting and monitoring incipient caries lesions.

The writing is excellent!

Sound scientific literacy and it is very easy to grasp the main concepts and hypothesis and methods.

The conclusion is well supported by the results and the discussion.

They use hypothesis methods to accept/reject the research questions.

After assessment of the manuscript "Evaluation of incipient enamel caries at smooth surface caries using SS-OCT", my recommendation is to accept with no further modifications.

Author Response

We wish to express our appreciation to the Reviewers for insightful comments on our paper. The comments have helped us significantly improve the paper.

Response to Reviewer 3

Comment 1.

After assessment of the manuscript "Evaluation of incipient enamel caries at smooth surface caries using SS-OCT", my recommendation is to accept with no further modifications.

Response: Thank you for supporting our study. We appreciate your educational comments.

Reviewer 4 Report

Dear authors,

Congratulations on your interesting study topic, which is an interesting one with wide applicability in dentistry.

Below you will find my recommendations.

Abstract - provide keywords because they are missing.

Introduction - I would suggest to move this paragraph  - We hypothesized (H0) that 3D image of intensity-based OCT would have no improvement of diagnosis for early enamel 74 caries compared with visual inspection based on ICDAS system. This null hypothesis was tested against the alternative hypothesis (H1) of difference. The sensitivity, specificity and area under the receiver operating characteristic curve (AUC) for the diagnosis of enamel caries were evaluated for SS-OCT and visual inspection based ICDAS system. Agreement with histology of SS-OCT and visual inspection were compared by weighted kappa. Obtained results were statistically analyzed at a significance level of α = 0.05. 7 - to the section - Materials and Methods.

I would add at the end of the the introduction the Aim of the study.

I also recommand if it is possible to restructure the informations from the introduction section, in order to be more clear and concreate.

Materials and Methods - please specify which were the standardized conditions used for digital photoghraphies.

I strongly reccomend to intoduce a section with conclusion, since they are in the text but are nor specified as subtitle. 

Author Response

We wish to express our appreciation to the Reviewers for insightful comments on our paper. The comments have helped us significantly improve the paper.

Response to Reviewer 4

Comment 1.

Provide keywords because they are missing.

Response: Thank you for pointing out. We added key words.

Comment 2.

Introduction - I would suggest to move this paragraph  - We hypothesized (H0) that 3D image of intensity-based OCT would have no improvement of diagnosis for early enamel 74 caries compared with visual inspection based on ICDAS system. This null hypothesis was tested against the alternative hypothesis (H1) of difference. The sensitivity, specificity and area under the receiver operating characteristic curve (AUC) for the diagnosis of enamel caries were evaluated for SS-OCT and visual inspection based ICDAS system. Agreement with histology of SS-OCT and visual inspection were compared by weighted kappa. Obtained results were statistically analyzed at a significance level of α = 0.05. 7 - to the section - Materials and Methods.

Response: Thank you for constructive and educational comments. We modified the paragraph and deleted “The sensitivity, specificity and area under the receiver operating characteristic curve (AUC) for the diagnosis of enamel caries were evaluated for SS-OCT and visual inspection based ICDAS system. Agreement with histology of SS-OCT and visual inspection were compared by weighted kappa. Obtained results were statistically analyzed at a significance level of α = 0.05.” from Introduction.

Comment 3.

I would add at the end of the introduction the Aim of the study.

Response: Thank you for pointing out. We added the Aim of this study in the last paragraph of Introduction.

Comment 4.

I also recommend if it is possible to restructure the informations from the introduction section, in order to be more clear and concreate.

Response: Thank you for constructive feedback. We modified Introduction and add a few sentences in 2nd paragraph and 3rd paragraph. References 5, 6 and 7 were also added.

Comment 5.

Materials and Methods - please specify which were the standardized conditions used for digital photoghraphies.

Response: Thank you for pointing out. We described the conditions for taking digital photograph.

Comment 6.

I strongly recommend to introduce a section with Conclusion, since they are in the text but are nor specified as subtitle

Response: Thank you again for constructive feedback. We added Conclusion in our manuscript.

Round 2

Reviewer 4 Report

Dear authors, congratulations for your overall work